# Understanding Cancer Survivors' Needs and Experiences Returning to Work Post-Treatment: A Longitudinal Qualitative Study

**Robin Urquhart [1,2,\*], Sarah Scruton [1] and Cynthia Kendell [3]**

[1]   Department of Community Health and Epidemiology, Dalhousie University, Halifax, NS B3H 1V7, Canada; sarah.scruton@dal.ca

[2]   Department of Surgery, Dalhousie University, Halifax, NS B3H 2Y9, Canada

[3]   Department of Surgery, Nova Scotia Health, Halifax, NS B3H 2Y9, Canada; cynthia.kendell@ccns.nshealth.ca

[\*]   Correspondence: robin.urquhart@nshealth.ca

**Abstract:** Background: This study aimed to understand Canadian cancer survivors' experiences during the return-to-work (RTW) process. Methods: A prospective qualitative longitudinal design was employed using the principles of phenomenological inquiry. Cancer survivors took part in three in-depth interviews: at the end of treatment, and 3 and 9 months after the first interview. Transcripts were analyzed using constant comparative analysis, guided by the Cancer and Work model. Results: A total of 38 in-depth interviews were conducted with 13 participants. The resultant themes were: (1) supports received or desired to enable RTW; (2) others' limited understanding of the long-term impacts of a cancer diagnosis and its treatment; (3) worries and self-doubts about returning to work; and (4) changing perspectives on life and work after cancer. Conclusions: Cancer patients returning to work after treatment often experience challenges throughout the process, including varying levels of support from others and a range of ongoing effects and motivation to RTW. There is a clear gap in terms of the professional supports available to these individuals. Future research should focus on investigating how to improve both quality and accessibility of supports in a way that is personalized to the individual.

**Keywords:** cancer; survivorship; return to work; qualitative methods

## 1. Introduction

Developments in the diagnosis and treatment of cancer mean that nearly two-thirds of Canadians diagnosed with cancer today will be long term survivors (i.e., they will live for more than 5 years after diagnosis) [1]. The increase in long-term survival means there is also an increase in the number of working-age individuals re-entering the workforce during or after treatment. In fact, past research has shown that 43–95% of survivors will return to work (RTW) within 12 months of diagnosis [2]. There are many reasons why an individual may choose to return to the workplace. Some describe it as an important piece of their self-identity and a way to showcase their accomplishments [3,4]. Others recognize it as an important source of socialization and friendship [5]. Though RTW is often necessary for financial reasons, research has shown that it is also a means of returning to normality, providing daily structure, and offering a distraction from an individual's diagnosis [4]. RTW is considered an important part of cancer recovery and many healthcare providers agree that it contributes to quality of life [6,7].

Despite the potential benefits of RTW, unemployment is higher, and productivity, earnings, and work retention are worse for cancer survivors who RTW compared to persons without a cancer history [8–18]. These outcomes are likely to be a result of the way in which cancer and subsequent treatments can affect work functioning. Many survivors endure physical and/or psychological side effects during their recovery, which can last for

years after completing treatment [19–21]. Examples of prevalent long-term effects include extreme fatigue, cognitive deficits, depression, anxiety, and physical limitations [22–24]. These challenges differ depending on the type of cancer and type of treatment received. Unsupportive work environments, lack of resources and guidance, discrimination, and poor job security may also impede the RTW process [3,4,25]. While some RTW programs and resources exist to aid survivors in regaining the strengths and skills needed to perform their jobs, there are mixed reviews about the effectiveness of these programs, and some are unsuccessful [26–28]. In fact, a Cochrane review found few interventions were effective in enhancing RTW outcomes in cancer patients/survivors [29]. In addition to the lack of professional programming, survivors often feel they are left without proper guidance from their employers or healthcare teams [3,25,26]. Without these supports, survivors are more likely to lose motivation or re-evaluate the importance of work [26].

An improved understanding of the transition process, specific to cancer survivors' needs and experiences, is critical to developing appropriate strategies to support cancer survivors during their RTW and to optimize RTW outcomes. The objectives of this study were to explore (1) cancer survivors' needs related to RTW and how these changed over time and (2) their experiences reintegrating into the workplace and what helps and hinders this process. Our aim was to identify strategies, policies, and practices that can support cancer survivors' RTW after cancer treatment. To our knowledge, this study is the first of its kind in Canada to longitudinally analyze cancer survivors' experiences during the RTW process for multiple cancer types and across multiple provinces. Previous studies within Canada have focused on either one specific cancer type, such as breast or osteosarcoma, or have included only one province [5,6]. Similarly, none have used a longitudinal design with serial interviews and therefore cannot reveal how experiences and perspectives change over time.

## 2. Materials and Methods

### 2.1. Study Design

This study employed a prospective qualitative longitudinal design and was underpinned by the principles of phenomenological inquiry. Phenomenology intends "to understand the phenomena in their own terms—to provide a description of human experience as it is experienced by the person herself" [30]. Qualitative longitudinal research explores lived experiences of change (or stability) over time and is an important means by which to study how people experience, interpret, and respond to change [31,32]. Feuerstein et al.'s Cancer and Work model guided study design and data collection [33]. This model conceptualizes work-related issues for cancer survivors through the concepts of health, functional status, work demands, work environment, and policy, procedures and economic factors. Ethics approval was obtained from the Nova Scotia Health Authority Research Ethics Board.

### 2.2. Participant Recruitment

Study participants were cancer survivors recruited by distributing study information to cancer clinics and community-based organizations and networks from across Canada, as well as social media (Facebook and Twitter). Participants contacted the study team directly if they were interested in participating. Eligibility criteria were as follows: aged 18 years and older, employed at the time of diagnosis, able to speak and understand English, had completed primary treatment (e.g., surgery, radiotherapy, and/or chemotherapy), and had an intention to RTW. Survivors of all cancer sites were eligible. Those with active disease post-primary treatment were eligible if their cancer was being managed as a chronic condition. Informed consent was obtained from each participant by a Research Associate prior to data collection.

*2.3. Interviews*

Data collection occurred from 18 March 2016 to 10 May 2017. Participants were invited to take part in three in-depth, semi-structured phone interviews: at the end of treatment, and 3 and 9 months after the first interview. The interview questions were created using the guidance of Patton and Rubin and Rubin [34,35], and based on the study objectives and the conceptual model. In addition to basic demographic questions, open-ended questions sought to understand participants' needs and expectations for RTW (Interview 1), and their experiences transitioning to work after treatment and views on critical supports and processes involved in the transition process (Interviews 2 and 3). If participants had not returned to work in subsequent interviews, the interview guide (Document S1) was adapted to explore their intentions, needs, and experiences with respect to RTW and how these evolved over time. The interviewer was a Research Associate, trained in qualitative methods and supervised by an experienced qualitative researcher (RU). Each interview was audio-taped and transcribed verbatim.

*2.4. Analysis*

A number was allocated to each participant and identifying information was removed prior to analysis to protect anonymity. Transcripts were analyzed using constant comparative analysis [36], guided by the Cancer and Work model [33]. Two approaches were used for the analyses to provide a rich understanding of the processes and changes which participants experienced [37]. First, individual accounts were analyzed longitudinally to examine the data in terms of individual narratives and how individuals experienced RTW. Second, accounts were compared and contrasted between participants to gain a broader understanding of their reintegration experiences, and how survivors' needs, experiences, and adjustments changed over time.

The transcripts were coded and categorized, with the assistance of NVivo 12. To maximize rigor, the first two interviews were coded independently by two researchers (RU, SS) to confirm consistency. This led to the development of a preliminary coding scheme. One researcher (SS) then coded the remainder of the transcripts, adding codes as necessary. Emergent codes and resulting themes were identified, discussed, and refined by the research team through multiple discussions as the analyses ensued.

## 3. Results

A total of 38 in-depth interviews were conducted with 13 participants. One participant took part in only two interviews. The average time for the first, second, and third interviews were 36:38, 21:00, and 21:58 min, respectively. Participants resided in four Canadian provinces: Nova Scotia, Ontario, Quebec, and Alberta. Table 1 presents participant characteristics. The analysis resulted in four themes, which are presented below. Table 2 presents the key findings as they relate to each interview timepoint and study objective.

**Table 1.** Participant characteristics (*N* = 13).

| Characteristic | *n* | % |
|:---:|:---:|:---:|
| Age | | |
| ≤40 | 5 | 38.5 |
| 40+ | 8 | 61.5 |
| Gender | | |
| Man | 3 | 23.1 |
| Woman | 10 | 76.9 |
| Non-binary | 0 | 0.0 |
| Marital Status | | |
| Partnered | 8 | 61.5 |
| Single | 5 | 38.5 |
| Children or Dependents | | |
| Yes | 7 | 53.8 |
| No | 6 | 46.2 |
| Place of Residence | | |
| Rural | 3 | 23.1 |
| Urban | 9 | 69.2 |
| No Answer | 1 | 7.7 |
| Job Category | | |
| Business, Management, & Administration | 4 | 30.8 |
| Engineering, Manufacturing, & Technology | 2 | 15.4 |
| Health Science Technology | 4 | 30.8 |
| Human Services | 3 | 23.1 |
| Pre-Diagnosis Work Hours | | |
| Part Time | 3 | 23.1 |
| Full Time | 10 | 76.9 |
| Cancer Type | | |
| Blood | 2 | 15.4 |
| Brain | 1 | 7.7 |
| Head and Neck | 2 | 15.4 |
| Breast | 5 | 38.5 |
| Ovarian | 2 | 15.4 |
| Abdominal | 1 | 7.7 |
| Cancer Treatment * | | |
| Chemotherapy | 10 | 76.9 |
| Surgery | 10 | 76.9 |
| Radiation | 7 | 53.8 |
| Hormone Therapy | 3 | 23.1 |
| Stem Cell Transplant | 1 | 7.7 |
| Returned to Work ** | | |
| Yes | 11 | 84.6 |
| No | 2 | 15.4 |

* Many participants received multiple cancer treatments. Therefore, many participants are represented in more than one row and this column does not equate to 100%. ** Return to work status at the third interview.

**Table 2.** Key findings as they relate to interview timepoints and study objectives. T1 = timeline 1; T2 = timepoint 2; T3 = timepoint 3.

| Interview | T1: End of Treatment | T2: 3 Months Post-Treatment | T3: 9 Months Post-Treatment * |
|---|---|---|---|
| **Study objective** | Explore needs related to return to work (RTW) | Explore ongoing needs and how these change over time<br>Explore experiences reintegrating back to work<br>Identify what helps and hinders the RTW process | |
| **Key findings** | Needs at the end of treatment include information about RTW, navigating the RTW process, formal psychosocial support, and employee accommodation | Ongoing needs include formal psychosocial supports and employee accommodation<br>Prevalent ongoing late effects include mental health issues, fatigue, pain, neuropathy, and cognitive impairment<br>High levels of emotional and practical support upon initial return to work, which decline over time<br>A limited understanding by others (family, friends, employers, co-workers, and/or members of healthcare team) of the ongoing late effects and the impacts on work<br>Worries and self-doubt can surface due to unforeseen challenges returning to work, including an inability to work at pre-cancer levels<br>Key enablers to RTW can include RTW navigation, peer supports, formal psychosocial services, and direct support from employers such as increased communication and flexibility<br>Many people develop new perspectives on life and work after a cancer diagnosis and treatment | |

* Eleven of 13 participants had returned to work at 9 months post-treatment.

*3.1. Supports Received or Desired to Enable RTW*

All participants experienced high levels of emotional support from their family and friends with respect to RTW, and many discussed how their families/friends worried that they were rushing their return. One participant described it this way:

> "They think I'm crazy. They think I should, ahh, be relaxing and trying to get my strength back . . . They're fine with me going back a little bit, but they think that I've gone back too much. Like that I'm doing too much now." (P8)

Similarly, all but one participant expressed high levels of moral and practical support from their employers and co-workers during their diagnosis and treatment periods, and upon their initial RTW. Ten participants had the opportunity to complete a gradual RTW process rather than return immediately full-time. Participants also described how many of their employers encouraged them to take breaks when needed and gave them flexibility in terms of workload. As one participant stated:

> "In the beginning, he (supervisor) told me, you know, 'Listen, you want to stop? Take a break. Go outside. Take some fresh air.' Because, he could see sometimes in my face that, ahh, I wasn't really feeling so good. That was early beginning when I was doing my first two-hour shift." (P4)

Participants discussed a range of experiences in terms of the supports they received from their care teams and insurance companies regarding RTW. While some participants felt they received adequate support from their insurance company or cancer care team, many discussed having to find supports themselves, or that the supports they did receive were suboptimal in terms of preparing them for their RTW journey. For example, many participants noted that they did not receive any referrals to formal psychosocial support from their care team, and they were not informed of specific supports available to navigate the RTW process by their care team or insurance company. Most also discussed how their

cancer care team did not discuss RTW as a part of their care and management, or discussed this issue in a rushed way:

> "The only person I really see on the cancer team is my doctor and a nurse and nobody had explained to me that that's kind of how it works. Maybe because they didn't know or anything like that. So, I don't feel like I've gotten a lot of support from them. I've had to do the work myself and granted I can recognize that they're probably pretty busy, umm, but still." (P2)

Supports that helped their RTW process included meeting other survivors in similar circumstances, prioritizing self-care, engaging in physical activity and other lifestyle programs, participating in hospital-based services (e.g., drop-in sessions, support groups), accessing psychological services through their employer, and receiving accommodation from their employer, including a gradual RTW. A minority of participants did discuss how their primary care provider was a source of both moral and practical support along their RTW journey instead of their cancer care team, reassuring individuals that they were able to RTW or advocating on their behalf if they needed more time away from work:

> "I hadn't talked to the cancer team. I had talked to my family doctor and, ahh he, you know, we discussed it and I said, 'you know what? I'll go back and try it and if I find it's too much for me well, you know, it will be too much.' He was a great supporter and felt that, you know, like I could do it because I seemed to be okay." (P6)

However, other participants noted that their primary care providers lacked an understanding of the unique side effects from their cancer treatment and how these impact RTW. As one participant described:

> "I feel like my family doctor is also putting pressure. Like, he doesn't really fully understand brain fog and fatigue and, you know, just the way he deals with what I say, he just, he doesn't understand that these are such serious side effects of cancer and he tells me that they are just short-term and if I go back to work, and once I get back in the routine of things, I'll be fine." (P1)

When asked, participants described a number of supports they felt would have helped with the RTW process: dedicated staff/navigator to assist with insurance companies, speaking with a peer who had similar experiences, formal psychosocial supports, and more support directly from their employer such as increased communication and flexibility during their return. One participant said:

> "I think it would be helpful if there was someone that people could, you know, like specifically that people could go talk to about transitioning out of, you know, being under full-time care all of the time and doctors every day to, you know, just being back in limbo and it's hard to put it out of your head that, you know, now it's just a wait and see game." (P8)

### 3.2. Others' Limited Understanding of the Long-Term Impacts of a Cancer Diagnosis and Its Treatment

For most participants, the feeling that family, friends, employers, and/or co-workers did not understand what they were going through was a pervasive experience. This related specifically to the existence of ongoing, long-term effects of their cancer and/or its treatment and the impact these effects had on RTW and remaining in the workforce. As one participant said:

"You can't really explain it to people because it doesn't, ahh, they just go, 'oh, I understand.' But they don't. You don't really understand that your brain just doesn't work the same anymore. It doesn't do what it did before. It's very, it's like having a stutter in your brain. And, ahh, just very tiring and you know. And, when I say, 'oh I'm tired' at the end of the day everybody just says, 'oh well maybe you shouldn't be doing so much' and it's like well, who is going to pay the bills if I don't, you know, do it." (P8)

In fact, all participants experienced long-term effects that were invisible to others. The most prevalent was a high level of fatigue, which was experienced by all participants. Other invisible long-term effects common to participants were fear of recurrence, anxiety and other mental health challenges, neuropathy, pain, and brain fog (e.g., memory loss, attention issues, and other cognitive deficits). For a substantial minority, this resulted in experiencing pressure from family and friends to RTW earlier than they felt ready to return. The lack of understanding from others impacted their emotional health and added to the challenges of returning to work after treatment:

"So, I think as a society, there's this assumption that say, you know, I've finished chemotherapy and now I'm healthy and I'm ready to go back to work. I think there's not . . . There needs to be this understanding that once you get diagnosed with cancer you are dealing with your health for the rest of your life and that becomes your priority. And it's not possible for everyone to go back to full-time work. For some people it could take years. Some people never get to go back." (P1)

Approximately half of participants felt pressure to RTW, either from their physician, employer, or insurance company, before they felt ready to re-enter the workforce. Those who worked as healthcare professionals described an added layer of vulnerability in that they felt required to RTW to maintain their certification or licensure. Several participants also discussed pressure to RTW due to a lack of job security. As a result of these pressures, participants often described a lack of control around RTW, and many experienced a swift RTW that led to them feeling like they had returned too early, or having to leave the workplace for a second time.

"I just feel it was too much, because, you know, I wish I had that opportunity to stay home and go back later, but the type of job, it was no job security and if you weren't there somebody else would have been hired. So, I sort of, you know, pushed myself to go, but now if ever I give anybody advice it's make sure your treatment's over and you're mentally and physically ready to go back. That would be my thing." (P6)

Despite experiencing high levels of emotional support from co-workers upon their initial RTW, a substantial minority also reported declining levels of support as the time from end of treatment progressed. They sensed that others saw them as cancer-free and healthy, and increasingly felt they should be 'back to normal.' One participant described this as:

"I think it would be nice if people understood that even though you look okay, what's going on in your brain isn't always okay because you're dealing with a lot of umm unknowns and if people . . . I find that people have assumed because my hair is growing back and I have eyebrows that I look so healthy and then they don't know what's going on with your emotional state." (P7)

*3.3. Worries and Self-Doubts about Returning to Work*

Most participants expressed concerns that they would be and/or were unable to work at the same level as they did before their cancer diagnosis. For the most part, this related to feeling unable to work full days due to ongoing fatigue, pain, and/or cognitive impairment.

In fact, many participants discussed feeling unprepared for how challenging their ongoing effects would be and how these effects would impact RTW in both the short- and long-term.

> "I consented to treatment of, ahh, like long-term damage to healthy brain tissue because they radiate not only the abnormal cells within the margins, but ahh it affects the sort of healthy brain tissue cells. So, one of my concerns is, ahh is umm, how does that affect work performance in the long-term? What sort of work, ahh, will I be able to do umm 10 or 20 years from now? When they talk about cognitive deficits like how does that, how does that actually manifest itself? What does that mean? What does that look like umm in like a 9 to 5 sort of regular work day context?" (P3)

For some, unique issues predominated their worries. For example, a participant who worked in health care worried that going back to the worksite would trigger unpleasant memories and reactions regarding their cancer treatment. Many participants described their worries as lowering self-esteem as it related to work, compared to before their diagnosis.

> "Like, I don't want to go back to work unless my mental health is good enough to and so, I just feel like those are things that no one really understands and I kind of just need to defend myself all the time. It's almost like I'm almost fighting to like tell people I'm not well enough to go back to work and that then affects my self-esteem, because I feel like I should be well enough but I'm not." (P1)

Importantly, most also worried about the long-term effects of their cancer and its diagnosis on their work performance, including if they experienced a recurrence. For most participants, their ongoing side effects and resultant concerns led to a loss of motivation to RTW, a loss of identity, ongoing guilt, and other mental health challenges that further impeded their ability to RTW in healthy and productive ways.

> "I know for me, like being home, after like working all the time. Like, not having a sense of meaning kind of in my life a little bit that I got from work before. Umm that was like, I think a big challenge and like so much of my identity was tied up in my work. Umm and like my profession, my relationship with my colleagues and all that kind of stuff, and so, when you take that away umm like it makes me question like, who are you? Like, what is your identity? . . . I think that our society and our culture values work so much in helping establish who we are as an individual and our relationship with other people that umm when you don't have work, what do you have to replace that? . . . having to help rebuild that sense of self and identity again, I guess, is like a struggle." (P2)

Finally, a number of participants described situations whereby their employer doubted their capabilities and worried that their cancer diagnosis meant they were unable to work at their pre-cancer level or carry out the same duties/tasks. This led participants to feel their employer was excessively observing their performance or that they were being discriminated against in their workplace. "I am concerned if I do end up with an accommodation that I'm going to be treated differently, because people are going to say, you know, that job should have gone to someone else. I don't know how people treat people who've been accommodated." [P13] Some participants worried they might lose their jobs or employee benefits if they did not return promptly and in a full-time capacity.

Despite the barriers faced, all participants were highly motivated to re-enter the workforce for a multitude of reasons. Though financial pressure was most commonly mentioned as a reason to RTW, participants also sought social interaction, routine, and distraction from their diagnosis.

*3.4. Changing Perspectives on Life and Work after Cancer*

Most participants felt they had changed in important ways since their cancer diagnosis, which provided them with a new perspective on life and work. These new perspectives were described in at least three ways. First, many participants, especially those under

40 years of age, described considering career changes and wanting a career that was more meaningful than the one they currently had.

> "I just want to do something that I find some enjoyment in and I sort of feel the need to help at this point, to do something. So many people have helped me I kind of want to, I know it's sound cliché, but I feel like I want to just sort of give back for everyone who has helped me." (P14)

Second, participants discussed the importance of self-care and how they realized that prioritizing one's own care, through activities like exercising more or taking more time for oneself, was important to their overall health and well-being. In this way, taking time away from work was an 'okay' thing to do. "It's just really listening to myself. When I wake up in the morning, if I feel horrible I just tell my employer and that's it. . . . I don't feel as guilty, I guess, umm listening to what I need." (P10)

Finally, participants discussed how, when faced with their mortality, they realized they cannot take things for granted. That is, they cannot believe life or work will always be available or the same, and therefore need to appreciate the things they have in the moment.

> "The diagnosis of a brain tumor is a life changing umm event . . . when it happens, it kind of shatters your illusions of umm sort of what you are doing. You are faced with your own mortality. Umm yes, I think people are, they are cognizant of it every day. It's just umm they take it for granted, the fact that you are alive and healthy and you don't really umm you don't really clue in to the fact that your days are numbered and that you are a mortal . . . For me, it's more umm especially with the radiation therapy. I don't know what effect that will have in umm the long-term." (P3)

## 4. Discussion

This study sought to understand participants' needs and experiences returning to work after a cancer diagnosis and subsequent treatment, and whether and how these experiences change over time. The analysis revealed that the RTW process often follows a non-linear pathway that varies between individuals. Despite the multitude of challenges faced, most participants did RTW by the end of the study period. The majority reported high emotional support from family, friends, employers, and co-workers, but levels of support were lower from cancer care teams and insurance companies. Participants felt that conversations about RTW with their physicians were often rushed, or that their physicians did not have a solid understanding of their ongoing effects and how these impacted their ability to RTW. Consequently, many experienced a lack of referral to important resources that they deemed would have been helpful, such as psychosocial supports or assistance in navigating the RTW process. Many participants felt that, in general, there is a lack of understanding of the long-term impacts of cancer. Due to many of these side-effects being invisible, participants discussed how others often assume they are improving physically or emotionally, leading to diminishing support over time. Most participants experienced worry and self-doubt regarding their ability to work at their pre-treatment level, mostly in relation to high levels of fatigue and the inability to get through a full workday. Finally, many participants experienced a shift in perspective, with a heightened awareness of their own mortality and the importance of life outside of work.

This study was the first of its kind in Canada to longitudinally assess the RTW process as people transition from active cancer treatment to life beyond cancer. The longitudinal nature allowed us to understand how experiences change or evolve as survivors progress through their RTW. Previous studies have reported that social support from employers and co-workers was a significant factor associated with a successful RTW [26]. However, our study revealed that this support often wanes over time as the participants appear healthier on the outside, regardless of how they feel physically or emotionally. This decrease in support could also be due to employers' or co-workers' lack of experiences dealing with a cancer survivor's RTW, as noted by previous studies. Tiedtke et al. found that employers

have difficulties balancing the interests of both the business and the employee, and they are unsure what their role is in the process [25,38]. Eguchi et al. found that co-workers often have similar difficulties, and a lack of experience working with a cancer survivor can be an obstacle to a successful RTW [39]. These findings suggest that continued emotional support from social contacts, such as friends or co-workers, during the long-term recovery is valuable and appreciated. Education for employers and co-workers around the RTW process for cancer survivors may be warranted to facilitate long-term recovery.

Participants in this study felt that both their primary care and oncology care teams were often unprepared or unable to have conversations about RTW due to time constraints, an under appreciation of their ongoing effects, or a lack of understanding of the RTW process. Similar findings have been reported in previous studies both in Canada and other parts of the world, which demonstrate that physicians have an important role in this process that may be overlooked [5,7]. Our study revealed specific supports that participants felt would have been helpful during their RTW. These commonly included easier access to psychosocial supports, speaking to a peer with a similar experience, or having access to a support person to navigate the insurance process. These may be target areas for cancer survivorship care or future research.

All but one participant identified fatigue as a major barrier to returning to work, which is in line with previous studies [19,20,23]. This was often accompanied by doubt in their abilities to perform their job at a pre-diagnosis level, as well as worry regarding long-term effects or recurrence. Common reasons for returning to work, aside from financial pressure, included seeking social interaction, routine, and distraction from their diagnosis. This supports others' findings that RTW is an important part of the cancer recovery process, as it allows survivors to return to a sense of normalcy or regain an important piece of their identity [4]. Each participant experienced their own unique path to RTW, and individual needs differed from person to person. This provides further evidence to the suggestion of Wells et al. that RTW supports should follow a person-centered approach that acknowledges individual differences [3].

This study has several limitations. First, it is possible that the participants who chose to take part in our study were highly motivated to RTW or had favorable work circumstances. Second, all but three participants received chemotherapy as part of their treatment, which has been linked to a more difficult time returning to work, and most participants were women [26,40]. Thus, our findings may not reflect the experiences of those who are less motivated to RTW or who face substantial barriers, or of men as they RTW after treatment. Indeed, we know that the gendered nature of work results in differences in work quality and income as well as the precariousness of work settings, which may impact RTW experiences after a cancer diagnosis and treatment [41,42]. Despite these limitations, this study provides us with a rich understanding of survivors' lived experiences of their RTW pathways and the meaning ascribed to these experiences over time. The longitudinal design provides an important contribution to the growing area of RTW research.

Future research should include investigation into the supports that participants identified as essential to an optimal RTW experience. This may include the creation of positions to guide patients through the RTW process and interventions for physicians and employers to enhance role clarity and knowledge around RTW processes. Research should focus on understanding how RTW interventions and supports can be personalized to each individual's unique needs and circumstances. Additional longitudinal research would continue to provide unique insight into how survivors' needs and experiences change over time.

## 5. Conclusions

Cancer patients returning to work after treatment often experience challenges throughout the process. Although survivors receive varying levels of support and experience a range of ongoing effects and motivation to RTW, our analysis revealed four main themes across the dataset: (1) there are important supports that enable RTW, (2) family, friends, employers, and/or co-workers often do not see and understand the long-term impacts of

cancer diagnosis and its treatment, (3) many survivors experience worry and self-doubt about returning to work, and (4) many change their perspectives on life and work after cancer. There is a clear gap in terms of the professional and personal assistance available to these individuals, and future research should focus on investigating how to improve both quality and accessibility of supports in a way that is personalized to the individual.

**Supplementary Materials:** The following supporting information can be downloaded at: https://www.mdpi.com/article/10.3390/curroncol29050245/s1, Document S1: Interview guide.

**Author Contributions:** Conceptualization: R.U. and C.K.; Methodology: R.U.; Formal Analysis: R.U. and S.S.; Writing—Original Draft Preparation: R.U.; Writing—Review & Editing: S.S. and C.K.; Funding Acquisition: R.U. All authors have read and agreed to the published version of the manuscript.

**Funding:** This research was funded by a 2014 New Investigator Award from the Beatrice Hunter Cancer Research Institute.

**Institutional Review Board Statement:** The study was conducted according to the guidelines of the Declaration of Helsinki, and approved by the Research Ethics Board of Nova Scotia Health (protocol number 1018267, approved 18 December 2014).

**Informed Consent Statement:** Informed consent was obtained from all subjects involved in the study.

**Data Availability Statement:** The data presented in this study are available on request from the corresponding author. The data are not publicly available due to privacy and confidentiality considerations.

**Acknowledgments:** We gratefully acknowledge all study participants, who took their time to participate in this study. We are also grateful for the support of Emily Drake, Research Associate, who helped recruit participants and conducted the interviews, and Margaret Jorgensen, Project Coordinator, who assisted with study operations.

**Conflicts of Interest:** The authors declare no conflict of interest. The funder had no role in the design, execution, interpretation, or writing of the study.

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
