# Peer review of "Understanding Cancer Survivors’ Needs and Experiences Returning to Work Post-Treatment: A Longitudinal Qualitative Study"

_curroncol, doi:10.3390/curroncol29050245_

Round 1

Reviewer 1 Report

Thank you for the invitation to review this manuscript. The topic is highly relevant and so many unknowns remain on Canadian cancer survivors' experiences with returning to work (RTW). There is even less published on the longitudinal RTW experience of these Canadian cancer survivors which makes the contribution of this paper to the literature even more valuable.  

Line 48: I would not be so inclined as to say that RTW programs exist to aid cancer survivors more like "some" programs exists. When RTW program exists, few are specified or adapted to fit the unique needs of cancer survivors.

2.2 from the 38 participants who took part in the study, can you list how many were recruited via Facebook and Twitter versus from cancer clinics and community-based organizations?

2.2: is there any reason that study participants were not interviewed from the time of treatment rather than after completing treatment?

2.3 the dates seem to be in a different font (line 94).

2.3 can you add the length of time on average each interview 1, 2, and 3 lasted?

2.4 line 110: please add a reference to the constant comparative analysis.

Table 1: do you have information on their education level? If so please add. As we know, higher education does lead to higher return to work rates. In the table, the return to work row yes and no, is that to represent their status at 1 year? if so, please specify it in the table. 

Line 291: is it possible also that cancer survivors return for fear of losing their job? it's not listed but it is a common reason usually amongst the others listed in the paper. 

Overall under the results section, I would recommend providing fewer quotes or shortening the quotes and more interpretation of the data. With so many quotes and little text, it feels like too much unanalyzed raw data which is left to the reader to interpret. Interpreting the data does not mean discussing it as you would under discussion, but providing the reader with the interpretation of the output of your data. The output being the raw verbatim accounts. 

If possible, and considering the strength of this study being longitudinal, I would add a timeline figure to represent the study objectives (needs to RTW and change over time, and reintegration of the workforce, and facilitators and barriers). See the paper by Bilodeau et al., 2019 as an example of a RTW timeline figure. 

Line 330: is it possible to add the timeline of RTW since the end of treatment like the majority returned 6 months following the end of their treatment?

332: is it accurate to say that support was appropriate in the initial return but then dwindle as time went by? That is what I seem to have understood from some of the results/quotes presented?

Line 348: Here is the first time we read about the term RTW "pathway". Could you please say more about what is meant by pathway and was that a focus of the study?

Line 407: instead of having "others" please identify who these others are.

Author Response

Comment 1: The topic is highly relevant and so many unknowns remain on Canadian cancer survivors' experiences with returning to work (RTW). There is even less published on the longitudinal RTW experience of these Canadian cancer survivors which makes the contribution of this paper to the literature even more valuable.  

Response: Thank you.

Comment 2: Line 48: I would not be so inclined as to say that RTW programs exist to aid cancer survivors more like "some" programs exists. When RTW program exists, few are specified or adapted to fit the unique needs of cancer survivors.

Response: We agree entirely with the Reviewers’ point. We have changed this line to state that “some” programs exist.

Comment 3: 2.2 from the 38 participants who took part in the study, can you list how many were recruited via Facebook and Twitter versus from cancer clinics and community-based organizations?

Response: Unfortunately, we cannot provide this information because we didn’t collect it at the time of initial contact. Upon seeing the study information (regardless of source), participants were instructed to contact research staff if they were interested and the process ensued from there. Nevertheless, as social media is used more often for research purposes (recruitment, dissemination), this is valuable information and data our research program will collect in the future.

Comment 4: 2.2: is there any reason that study participants were not interviewed from the time of treatment rather than after completing treatment?

Response: We chose to interview individuals upon their completion of treatment rather than from the time of treatment since we wanted to capture individuals at a time when the intensity of treatment (physical, emotional, and time intensity) was behind them and most were at a point when they could more easily contemplate/plan for life after treatment. Having said this, we recognize many individuals return to work while still on treatment, for various reasons.

Comment 4: 2.3 the dates seem to be in a different font (line 94).

Response: Thank you. We have ensured the font is the same throughout the methods section.

Comment 6: 2.3 can you add the length of time on average each interview 1, 2, and 3 lasted?

Response: Thank you for this question. The average time for the first, second, and third interviews were 36:38, 21:00, and 21:58 minutes, respectively. We have added the average length of time for each interview timepoint.

Comment 7: 2.4 line 110: please add a reference to the constant comparative analysis.

Response: Thank you for noting this oversight. We have added the following reference to the manuscript and have changed the ensuing citation numbers accordingly:

Strauss AL, Corbin JM: Basics of qualitative research: grounded theory procedures and techniques. Thousand Oaks, CA: Sage; 1990.

Comment 8: Table 1: do you have information on their education level? If so please add. As we know, higher education does lead to higher return to work rates. In the table, the return to work row yes and no, is that to represent their status at 1 year? if so, please specify it in the table. 

Response: Thank you for these questions. No, we did not collect education level (e.g., highest education obtained) and are unable to provide these details. The return to work row actually indicates their status at the third (final) interview. We have now specified this in the table.

Comment 9: Line 291: is it possible also that cancer survivors return for fear of losing their job? it's not listed but it is a common reason usually amongst the others listed in the paper. 

Response: This is a great question. Yes, four participants explicitly talked about their worry that they would lose their job or their employee benefits if they did not return to work in a prompt manner. We have now included additional statements to this effect in the Results section:

Those who worked as healthcare professionals described an added layer of vulnerability in that they felt required to RTW to maintain their certification or licensure.  Several participants also discussed pressure to RTW due to a lack of job security.

Some participants worried they might lose their jobs or employee benefits if they did not return promptly and in a full-time capacity.

Comment 10: Overall under the results section, I would recommend providing fewer quotes or shortening the quotes and more interpretation of the data. With so many quotes and little text, it feels like too much unanalyzed raw data which is left to the reader to interpret. Interpreting the data does not mean discussing it as you would under discussion, but providing the reader with the interpretation of the output of your data. The output being the raw verbatim accounts. 

Response: Thank you for this comment. The initial draft may reflect a style issue, in that we prefer to use the participants’ own voices where possibly rather than our own descriptions of their voices/experiences. Nonetheless, we certainly do not want to give the impression that there is too much unanalyzed data. We have revised the Results section with the Reviewer’s comment top of mind. Quotes have been shortened and additional description added as an output of our data analysis.

Comment 11: If possible, and considering the strength of this study being longitudinal, I would add a timeline figure to represent the study objectives (needs to RTW and change over time, and reintegration of the workforce, and facilitators and barriers). See the paper by Bilodeau et al., 2019 as an example of a RTW timeline figure. 

Response: We appreciate the Reviewer’s suggestion. Instead of a timeline figure per se, we have included a table (Table 2) that presents the key findings according to the interview timepoint and study objectives.

Comment 12: Line 330: is it possible to add the timeline of RTW since the end of treatment like the majority returned 6 months following the end of their treatment?

Response: See above.

Comment 13: 332: is it accurate to say that support was appropriate in the initial return but then dwindle as time went by? That is what I seem to have understood from some of the results/quotes presented?

Response: Yes, this is accurate. Most participants felt well supported upon the initial return but they perceived this support to decline as time went by – despite their ongoing effects and the impact of these effects on health recovery.

Comment 14: Line 348: Here is the first time we read about the term RTW "pathway". Could you please say more about what is meant by pathway and was that a focus of the study?

Response: Thank you for this comment. The terms pathway was not intended to reflect an actual pathway per se, but rather we were interested in studying the RTW process for individuals and what helped/hindered this process. We have removed the term pathway from the manuscript.

Comment 15: Line 407: instead of having "others" please identify who these others are.

Response: We have changed “others” to “family, friends, employers, and/or co-workers”.

Reviewer 2 Report

This is a methodologically solid contribution to understanding the difficult pathway of Returning-to-Work (RTW) after cancer in Canadian society. Although the sample is rather small (13 participants and 38 interviews), the paper employs an impressive longitudinal methodological approach that has a trans-local geographical coverage (these participants are based in 4 provinces from Canada). Another strong point of this research consists of a rigorous methodological procedure employed in coding and analyzing the qualitative data resulting from the serial interviews. Despite the fact that the findings are not spectacular (i.e., they do not depart from other findings reported in the existing scholarship on RTW), they are empirically grounded and methodologically valid. As such, they bring a contribution to the literature by consolidating the existing knowledge with a case-study from Canada. The authors also acknowledge the limitations of the research, especially the slightly disbalanced sample, where female cancer survivors working in urban areas are more prominent.

Based on these considerations, I would like to congratulate the authors and express my full support in favor of publishing this paper in Current Oncology.

Author Response

This is a methodologically solid contribution to understanding the difficult pathway of Returning-to-Work (RTW) after cancer in Canadian society. Although the sample is rather small (13 participants and 38 interviews), the paper employs an impressive longitudinal methodological approach that has a trans-local geographical coverage (these participants are based in 4 provinces from Canada). Another strong point of this research consists of a rigorous methodological procedure employed in coding and analyzing the qualitative data resulting from the serial interviews. Despite the fact that the findings are not spectacular (i.e., they do not depart from other findings reported in the existing scholarship on RTW), they are empirically grounded and methodologically valid. As such, they bring a contribution to the literature by consolidating the existing knowledge with a case-study from Canada. The authors also acknowledge the limitations of the research, especially the slightly disbalanced sample, where female cancer survivors working in urban areas are more prominent.

Based on these considerations, I would like to congratulate the authors and express my full support in favor of publishing this paper in Current Oncology.

Response: Thank you for your comments on our manuscript.